# The Marine *Catenovulum agarivorans* MNH15 and Dextranase: Removing Dental Plaque

**DOI:** 10.3390/md17100592

**Published:** 2019-10-18

**Authors:** Xiaohua Lai, Xin Liu, Xueqin Liu, Tian Deng, Yanli Feng, Xiaopeng Tian, Mingsheng Lyu, Shujun Wang

**Affiliations:** 1Jiangsu Key Laboratory of Marine Bioresources and Environment/Jiangsu Key Laboratory of Marine Biotechnology, Jiangsu Ocean University, Lianyungang 222005, China; huaxiaolai10@163.com (X.L.); liuxin1490@163.com (X.L.); xueqinliu123@163.com (X.L.); dengtian729@163.com (T.D.); ylfeng0@163.com (Y.F.); txpxiaopeng@163.com (X.T.); 2Co-Innovation Center of Jiangsu Marine Bio-industry Technology, Jiangsu Ocean University, Lianyungang 222005, China; 3Collaborative Innovation Center of Modern Biological Manufacturing, Anhui University, Hefei 230039, China

**Keywords:** *Catenovulum agarivorans*, dextranase, dental plaque, oligo-saccharides

## Abstract

Dextranase, a hydrolase that specifically hydrolyzes α-1,6-glucosidic bonds, has been used in the pharmaceutical, food, and biotechnology industries. In this study, the strain of *Catenovulum agarivorans* MNH15 was screened from marine samples. When the temperature, initial pH, NaCl concentration, and inducer concentration were 30 °C, 8.0, 5 g/L, and 8 g/L, respectively, it yielded more dextranase. The molecular weight of the dextranase was approximately 110 kDa. The maximum enzyme activity was achieved at 40 °C and a pH of 8.0. The enzyme was stable at 30 °C and a pH of 5–9. The metal ion Sr^2+^ enhanced its activity, whereas NH^4+^, Co^2+^, Cu^2+^, and Li^+^ had the opposite effect. The dextranase effectively inhibited the formation of biofilm by *Streptococcus mutans*. Moreover, sodium fluoride, xylitol, and sodium benzoate, all used in dental care products, had no significant effect on dextranase activity. In addition, high-performance liquid chromatography (HPLC) showed that dextran was mainly hydrolyzed to glucose, maltose, and maltoheptaose. The results indicated that dextranase has high application potential in dental products such as toothpaste and mouthwash.

## 1. Introduction

Dextran is a water-soluble polysaccharide linked by α-1,6-glucosidic bonds, which can be synthesized by bacteria such as *Leuconostoc mesenteroides, Streptococcus*, and *Rhizopus* [1,2,3]. Water-soluble exopolysaccharides of dextran mainly play a role in plaque formation [4,5]. In addition, microbes use sucrose to produce large amounts of dextran, which has led to a decrease in the yield and quality of sucrose in the sugar industry [6,7,8,9]. So far, several methods have been used to control the formation of dextran, such as antibacterial agents, mechanical removal, and biological enzymes [10,11]. It has been proven that dextranase can hydrolase dextran safety and effectively.

Dextranase (α-1,6-d-glucan-6-glucanohydrolase, E.C.3.2.1.11), an inducible enzyme, is a hydrolase that catalyzes the hydrolysis of α-1,6-glucosidic bonds in dextran [12,13,14]. Bio-enzymatic hydrolysis of dextran is not only used in the preparation of dextran and its derivatives, but also plays an important role in the production of beverages and syrups in the food industry [15]. In sugar industries, dextranase can hydrolyze dextran in sugar juice, which can reduce its viscosity and increase the yield of sucrose [8,9,16]. In medicine, dextranase can degrade the high molecular weight of dextran into a low molecular weight. The specific molecular weight of dextran has an antithrombotic effect and can be used as a blood substitute in emergencies [17,18]. Currently, dextranase produced by fungi is mainly added to oral care products such as toothpaste and mouthwash to treat oral biofilm [19,20,21,22,23]. Larsson et al. reported on the catalytic reaction of Penicillium dextranase, the crystal structure of the enzyme, and the product complex [24]. The optimum temperature of dextranase produced by fungi was higher than 50 °C and unstable under alkaline conditions [22]. It has also been reported that dextranase is mainly produced by terrestrial bacteria [25,26,27]. Cold-adapted enzymes from the ocean can serve as a source of new marine medicines [28,29] and have enormous biotechnology potential in dairy products, sugar, detergents, and bioremediation [30,31].

Dextranase produced by marine bacteria has the characteristics of alkali resistance, salt tolerance, and high catalytic activity at moderate temperatures [32,33,34]. Majeed et al. reported that alkaline dextranase may play an important role in the treatment of oral plaque because alkaline mouthwash products are more protective than enamel against acidic products [35]. Therefore, it is desirable to explore more marine bacterial dextranases. Dextranase with better characteristics is more suitable for oral care products and is still widely in demand for preventing dental caries.

In this study, we isolated marine bacteria producing dextranase; identified their physiological and biochemical characteristics and molecular biology; and then studied bacterial growth characteristics, dextranase properties, and the effect of plaque removal. All the results indicated that the characteristics of dextranase are excellent and worth studying to identify its further application.

## 2. Results

### 2.1. Screened Marine Strains Producing Dextranase

Five strains of dextran-producing strains were selected from blue dextran plates containing different marine samples. After comparing their hydrolysis abilities, the strain with highest enzyme-producing ability, MNH15 (Figure 1a), was selected (Table 1).

### 2.2. Identification of Strain MNH15

The morphological and biochemical characteristics of strain MNH15 are shown in Table 2. The strain was Gram-negative, aerobic, short rod-shaped (Figure 1b), grown at 37 °C, and did not grow at 4 °C. Using genomic DNA from strain MNH15 as a template, a 16S universal primer was used for polymerase chain reaction, and the amplified fragment was about 1.5 kb (Figure 2a). After the PCR product was recovered, it was ligated to the pMD19-T cloning vector and transferred to *E.*-*coli*-competent cells, after which the positive clones were selected and sequenced to obtain a 1595 bp sequence. The 16S rDNA gene sequence was retrieved by GenBank and compared with its 16S rDNA sequence, and the phylogenetic tree was calculated using MEGA software ver. 5.0 and Clustal ver. 1.83. The phylogenetic tree indicated that strain MNH15 belonged to the genus *Catenovulum* (Figure 2b). On the basis of the morphological and biochemical characteristics combined with gene sequence analysis, the selected strain was recorded as *Catenovulum agarivorans* MNH15.

The optimum growth temperature was 35°C (Figure 3a). The strain MNH15 could grow at a wide pH range, but a pH of 8.0 was optimum (Figure 3b). The strain was grown in a medium containing 1%–7% NaCl at an optimum concentration of 2% and did not grow without NaCl (Figure 3c). The logarithmic growth phase of the strain was 3–6 h, and the dextranase activity increased gradually between 24 h and 48 h (Figure 3d).

### 2.3. Dextranase-Producing Conditions

#### 2.3.1. Effects of Carbon and Nitrogen Source on Dextranase Production

The effects of different carbon sources and nitrogen sources on the dextranase-producing strain MNH15 are shown in Figure 4. Barley flour was the best carbon source for the production of dextranase. Soluble starch and maltose had a slight effect on the strain producing dextranase. However, potato starch, tapioca starch, lactose, corn starch, glucose, rice bran, bran, sucrose, and dextrin could promote the production of dextranase (Figure 4a). Casein was the best nitrogen source (Figure 4b), followed by soybean meal and peanut meal, whereas casein was costly and unsuitable for large-scale fermentation. Therefore, soybean meal was selected as the nitrogen source.

#### 2.3.2. Effects of Temperature, pH, and NaCl Concentration on Dextranase Production

The optimal temperature for producing dextranase from strain MNH15 was 30°C. When the temperature was higher or lower, the dextranase-producing capacity changed sharply (Figure 5a). The initial pH of the medium was adjusted to the range of 5.0–10.0, and the dextranase activity was measured after 48 h of fermentation. At a pH of 8.0, the strain was most beneficial to dextranase production (Figure 5b). When the NaCl concentration was 5 g/L, the dextranase-producing capacity reached its peak (Figure 5c). Moreover, the NaCl concentration had a significant effect on the growth and the enzyme-producing capacity.

#### 2.3.3. Effects of Inoculum Size, Aeration, and Inducer Concentration on Dextranase Production

As Figure 5d shows, the inoculum size had an important effect on dextranase production. The optimal inoculum amount was 3%. When 60 mL of medium was added to a 250 mL Erlenmeyer flask, the amount of aeration was most favorable for dextranase production (Figure 5e). The results clearly showed that the dextranase production needed an inducer (Figure 5f). A dextran T20 concentration of 8 g/L was the most beneficial to dextranase production.

### 2.4. Enzymatic Characterization

The dextranase was subjected to sodium dodecyl sulfate (SDS)-polyacrylamide gel electrophoresis (PAGE). The band in the Coomassie blue-stained gel was consistent with the position of the transparent band in the blue dextran gel (Figure 6). This result showed that the molecular weight of the dextranase was between 100 kDa and 130 kDa (around 110 kDa).

#### 2.4.1. Effects of Temperature and pH on Dextranase Activity and Stability

The effect of temperature on dextranase activity is shown in Figure 7a. Dextranase activity was very sensitive to temperature change, with the optimum temperature being 40°C. Dextranase activity peaked at a pH of 8.0, as shown in Figure 7b. The dextranase maintained a high activity level in the pH range of 6.0–8.0. In addition, the dextranase was very stable in the pH range of 4.0–9.0 at 25°C (Figure 7b). The thermal stability of dextranase showed that the residual activity level stayed at almost 100% following storage for 5 h at 30 °C (pH 8.0), and nearly 50% of the enzyme activity was lost following storage at 45°C for 5 h (Figure 7c).

#### 2.4.2. Effects of Metal Ions and Reagents on Dextranase Activity

The effect of metal ions on dextranase activity is shown in Table 3. In the presence of metal ions Sr^2+^, the dextranase activity increased from 100% (without adding compounds) to 128.71%, and the metal ions NH^4+^, Ni^2+^, Cd^2+^, Fe^3+^, Li^+^, Cu^2+^, and Co^2+^ had a strong inhibitory effect on dextranase activity. Metal ions such as Ca^2+^, K^+^, Zn^2+^, Mg^2+^, and Ba^2+^ did not have a significant inhibitory effect. Moreover, the main compounds in dental care products, such as sodium fluoride, xylitol, sodium benzoate, lauric acid, and ethanol, had no significant effect on dextranase activity. However, we found that 1 mM sodium dodecyl sulfate (SDS) had a certain inhibitory effect on dextranase activity (Table 4), which was similar to results reported by Ren Wei et al. [35].

#### 2.4.3. Substrate Specificity and Final Hydrolysis Products

The substrate specificity of dextranase was evaluated using materials with different glucosidic linkages (Table 5). Dextranase had high specificity for dextran containing α-1,6-glucosidic linkages. When the substrate concentration was 3%, the optimal substrate was dextran T500. However, the dextranase had poor hydrolysis ability for soluble starch, even though the soluble starch was composed of α-1,4-glucosidic bonds. Pullulan, chitin, cyclodextrin, and mannan are mainly composed of α-1,4-glucosidic bonds, and the dextranase was unable to hydrolyze them, either. High-performance liquid chromatography (HPLC) showed that maltoheptaose, glucose, and maltose were the main hydrolysates of dextranase (Figure 8). Moreover, the peak area of the hydrolysate detected by HPLC was quantified using Empower GPC software (Gel Permeation Chromatography—GPC) (Table 6). When the hydrolysis reaction time was extended from 0.5 h to 3 h, the amount of maltotriose and maltotetraose decreased slightly, which was similar to results reported in the literature [35,36]. This indicated that dextranase is an endotype dextranase.

### 2.5. Effect of Dextranase on Plaque

*Streptococcus mutans* is considered to be one of the important bacteria responsible for forming dental plaque and caries. To investigate the inhibition of dextranase on the biofilm formation of *S. mutans*, the minimally biofilm inhibitory concentration (MBIC) of this enzyme was determined (Table 7). The results showed that the experimental groups were significantly different from the control group, indicating that dextranase has a remarkable inhibitory effect on the formation of plaque. As the concentration of dextranase increased, the amount of plaque formed gradually decreased. When the dextranase concentration was 3 U/mL and 7 U/mL, the plaque inhibition rate was 52.3% and 91.79%, respectively.

The results of the scanning electron microscopy (SEM) showed that the structure of the plaque without the addition of dextranase was tight, and the extracellular polysaccharide and bacteria interweaved to form many channels (Figure 9). When the concentration of dextranase was continuously increased, the adhesion of bacteria decreased, the biofilm gradually became thinner, and the structure became looser. The micro-plate crystal violet staining combined with scanning electron microscopy showed that the dextranase significantly inhibited the plaque formed by *S. mutans.*

The biofilm inhibitory rate was calculated at an absorbance of 595 (A_595_) of the crystal-violet-stained biofilm without dextranase subtracted from A_595_ of the biofilm with dextranase, and divided by A_595_ of the biofilm without dextranase multiplied by 100%.

## 3. Discussion

Dental caries is a chronic disease affecting different population groups all over the world. Moreover, plaque supports bacteria in etching the teeth [17,32]. Dextranase can remove plaque, but it is still a challenge to find a dextranase that is suitable for use in oral products. Compared with terrestrial-derived biological enzymes, marine enzymes are more suitable for industrial applications because of their salt tolerance, low temperature resistance, and alkali resistance [33,34].

*Catenovulum* sp. is a new type of marine bacteria found in the marine environment in recent years, so there is little information on its properties, and there are few strains that produce dextranase. Compared with fungi and actinomycetes [27], bacterial fermentation production of dextranase has the advantages of a short fermentation time and a low temperature, which can save energy, manpower, and time in industrial applications. *C. agarivorans* MNH15 did not grow in the broth absence of NaCl, indicating that the strain is a halophilic bacterium that grows better at a pH of 8–9, and like *C. agarivorans* [37,38]. Compared with terrestrial bacteria, most marine bacteria are oligotrophic, and the enzyme-producing medium components are relatively easy to get, so *C. agarivorans* MNH15 has strong potential for application in industry settings [27,39,40].

The molecular weight of dextranase of bacteria was found to be between 60 and 114 KDa [27,41], and Koenig and Day found that the molecular weight of fungi was at least 23 KDa [42]. In contrast, we found that the molecular weight of dextranase produced by *C. agarivorans* MNH15 was about 110 KDa. The bacterium of the same genus, *Catenovulum* sp. DP03, was found to have a molecular weight of about 75 KDa [35]. Thus, further research on the structure of dextranase may contribute to a better understanding of biochemistry and biocatalysis. The optimal pH of dextranase was found to be 8.0. Moreover, the pH of the dextranase produced by most strains, such as *Arthrobacter sp.* Arth410 [43] and *T. pinophilus* H6 [27], was acidic. Our dextranase also had a lower pH sensitivity, indicating that the alkaline dextranase is suitable for mouth products [32,44,45]. The dextranase had good catalytic activity at a temperature range of 35 °C–45 °C, which is similar to human body temperature, indicating that it can be applied to the development of oral care products.

The activity of dextranase increased to 128.71% under the treatment of 5 mM SrCl_2_, which was consistent with the results reported by Wu et al. [17,39,45,46]. The same concentration of SrCl_2_ had little effect on the activity of dextranase produced by the same genus *Catenovulum* sp. DP03 bacteria, reported by Ren Wei et al. [35]. Moreover, MgCl_2_, NH_4_Cl, and NiCl_2_ only had a slight effect on dextranase activity, whereas CaCl_2_, BaCl_2_, KCl, and ZnSO_4_ had no significant effect on the activity of dextranase. The characteristics will benefit the application of oral care products [27]. The products of MNH15-dextranase-catalyzed substrate hydrolysis were mainly glucose, maltose, maltotetraose, and maltoheptaose, which was slightly different compared with other sources of dextranases [35,40,47,48]. The main hydrolysate of *Catenovulum* sp. DP03 dextranase was isomalto-oligosaccharide, which produces a small amount of glucose over time [35]. The hydrolysates of *T. pinophilus* dextranase reported by Yu-Qi Zhang et al. were isomaltose and a small amount of isomalto-oligosaccharide [27].

We found that several reagents commonly used in dental care products, such as ethanol, sodium fluoride, xylitol, lauric acid, and sodium benzoate, had no significant effect of the dextranase activity. The enzyme produced by marine microorganisms was a biologically active metabolite with excellent thermal stability, salt tolerance, and low temperature resistance [49,50,51]. The characteristics of dextranase imply an advantage when it is used in oral care products to remove plaque and prevent dental caries.

## 4. Materials and Methods

### 4.1. Samples and Chemicals

Sea mud, seaweed, and seawater samples were collected from Haizhou Bay in Jiangsu, China. Dextran (T20, T40, T70, T500) and blue dextran 2000 were obtained from GE Healthcare (Uppsala, Sweden). All other reagents were purchased from Sinopharm Chemical Reagent Corp. (Shanghai, China) and were of analytical grade.

### 4.2. Isolation of Dextranase-Producing Marine Bacterial Strains

We used a blue dextran plate to screen marine bacteria-produced dextranase. The screening medium plates contained 10 g dextran, 1 g yeast extract, 5 g peptone, 2 g blue dextran 2000, 20 g agar powder, and 1 L aged sea water, at an initial pH of 8.0. We diluted the sample suspension with an appropriate amount of sterile distilled water and spread it evenly on a blue dextran plate. After incubating it for 2 d at 25 °C, the strain producing dextranase was selected according to a transparent zone around the colony. The colony diameter and the diameter of the transparent zone of the target strain on the plate were measured, and the hydrolysis ability was determined by the square of their ratios. The selected strains were inoculated into the fermentation shake flask (excluding agar and blue dextran 2000), and after fermentation at 180 °C for 48 h at 25 °C, the dextranase activity was measured using a standard method, and the strain with the highest enzyme production rate was selected.

### 4.3. Identification and Characterization of Bacteria

The phenotypic characteristics of strain MNH15 were identified by observing colony morphology, light microscopy, and SEM. We then identified various physiological and biochemical reactions according to the Berger Bacterial Identification Manual (9th Edition, Baltimore: Williams & Wilkins). The genome of strain MNH15 was extracted with the bacterial genome extraction kit, and the 16S-rDNA gene was amplified with primers 27F (5′-AGAGTTTGATCCTGGCTCAG-3’) and 1492R (5′-GGTTACCTTGTTACGACTT-3’). The PCR product was gel-recovered, connected with pMD19-T vector (Takara Bio Inc. Japan), and transferred to DH5a-competent cells. The positive clones were then screened by blue and white spots. The extracted colony plasmids were verified using agarose gel and sent for sequencing (Sangon, Shanghai, China). The 16S-rDNA gene sequence was analyzed by BLAST analysis with the existing sequence in GenBank, and the phylogenetic tree was constructed using the neighbor-joining method to determine the taxonomic status of the strain.

After the seed was activated, the cells were cultured at different temperatures (0 °C–45 °C) to determine the optimal growth temperature. In order to prevent pH changes during the culture, different buffers (MES buffer, pH 5.0–6.0; PIPES buffer, pH 6.5–7.0; HEPES buffer, pH 9.0–11.0) with a final concentration of 10 mmol/L were added to the medium to maintain the pH range of 4.0–10.0, and the culture was carried out at the optimum temperature. The medium was prepared with a trace-salt solution (without NaCl), and the NaCl concentration ranged from 0 to 100 g/L. It was then cultured under optimum temperature and pH conditions. The logarithmic growth phase of strain MNH15 included inoculating the 3% seed solution, and the absorbance at a wavelength of 600 nm was measured every 3 h under optimal growth conditions. The optimal culture time for producing dextranase was in the range of 0 h and 72 h. After 12 h of culturing, samples were taken every 6 h, and the enzyme activity was measured under standard conditions.

### 4.4. Culture Conditions for Dextranase Production and Purification

The culture medium for producing dextranase comprised 5 g of yeast extract, 5 g of peptone, 10 g of dextran, and 1 L of aged seawater, at an initial pH of 8.0. After the strain was activated, the inoculated medium was cultured at 30 °C at 180 rpm for 48 h, the fermentation broth was centrifuged (10,000 rpm, 30 min, 4 °C), and the supernatant was taken to determine the dextranase activity. Then, the supernatant was filtered through a 0.45 μm membrane, ultra-filtered using a hollow fiber column, and a molecular weight of 30 KDa was cut off. After being purified, the freeze-dried powder was prepared using a freeze-drying machine (Labconco Corp., Kansas City, MO, USA).

### 4.5. Enzyme Assay

Then, 50 μL of the enzyme solution was added into 150 μL of 3% dextran T20 (50 mM, pH 8.0 Tris-HCl buffer), and kept at 40 °C for 15 min. The amount of reducing sugar was determined using the 3,5-dinitrosalicylic acid (DNS) method. The enzyme activity dictated that, under the above reaction conditions, the amount of enzyme required to catalyze the release of 1 μmol of maltose per minute was one unit of activity [52].

### 4.6. Conditions Needed for Producing Dextranase

#### 4.6.1. Effects of Different Carbon and Nitrogen Sources on Dextranase Production

The yeast extract and peptone in the fermentation medium were replaced by a common carbon source of 10 g/L and a common nitrogen source of 5 g/L, respectively. After being cultured at 30 °C, 180 rpm for 48 h, the supernatant of the fermentation broth was taken to determine the activity of dextranase.

#### 4.6.2. Effects of Temperature, Initial pH, and NaCl Concentration on Dextranase Production

The culture medium was incubated at different temperatures (15 °C–40 °C) for 48 h under the previously mentioned optimal medium. The initial pH of the fermentation medium (5.0–10.0) was adjusted to determine the optimal initial pH of the enzyme produced by the strain and incubated at 30 °C for 48 h. Different concentrations of NaCl were added to the medium, and the fermentation was carried out at an optimal temperature, pH, and time. The enzyme activity was then measured using a standard method.

#### 4.6.3. Effect of Inoculum, Aeration, and Inducer Concentration on Dextranase Production

The seed solution was inoculated into the fermentation shake flask according to different amounts (1%–6%). The different volumes of medium (20–90 mL) were then added in a 250 mL Erlenmeyer flask. Dextran 20000 was used as an inducer to induce the production of dextranase, and different concentrations of dextran (0%–1.6%) were then added. Then, we detected the dextranase of the supernatant.

### 4.7. Enzyme Preparation and In-Gel Activity Assay

The supernatant was lyophilized and stored at −20 °C. The molecular mass of dextranase was visualized through SDS-polyacrylamide gel electrophoresis (PAGE). The activity level of dextranase in the gel was determined by an 8% native PAGE gel containing 0.5% blue dextran.

#### 4.7.1. Effects of Temperature on Dextranase Activity and Stability

In order to evaluate the effect of temperature on the activity of the enzyme, the enzyme activity after incubation at 20 °C–60 °C was measured. The relative residual activity of dextranase was determined through a standard method after leaving the enzyme solution at different temperatures (30 °C, 40 °C, 50 °C) without adding substrate for 1–5 h to investigate the thermal stability of the enzyme. The percentage of the maximum enzyme activity indicated the relative activity.

#### 4.7.2. Effect of pH on Dextranase Activity and Stability

Determination of dextranase activity at different pH values ranging from 4.0 to 9.0 (acetate buffer, pH 3.0–5.5; phosphate buffer, pH 6.0–8.0; Tris-HCl, pH 7.5–9.0) and at 40 °C. To evaluate the pH stability of dextranase in the absence of substrate, the dextranase was placed in a buffer of pH 4.0–9.0 at 25 °C for 1 h. Using dextran 20000 as a substrate, we measured the residual viability of the enzyme under standard conditions.

#### 4.7.3. Effects of Metal Ions and Other Reagents on Dextranase Activity

Next, we measured the effects of the following metal ions and reagents on dextranase activity: Ca^2+^ (CaCl_2_), Ba^2+^ (BaCl_2_), Mg^2+^ (MgCl_2_), NH_4_^+^ (NH_4_Cl), Ni^2+^ (NiCl_2_), Fe^3+^ (FeCl_3_), Co^2+^ (CoCl_2_), Cu^2+^ (CuSO_4_), K^+^ (KCl), Zn^2+^ (ZnSO_4_), Sr^2+^ (SrCl_2_), Li^+^ (LiCl), Cd^2+^ (CdCl_2_), sodium fluoride, xylitol, sodium benzoate, ethanol, sodium dodecyl sulfonate, and lauric acid. The metal ions were set at two different concentrations.

#### 4.7.4. Substrate Specificity and Analysis of Final Hydrolysis Products

The substrate of the different glucosidic linkages was reacted with dextranase at 40 °C for 15 min, and the enzyme activity was measured. The relative enzyme activity was expressed at 100% of the maximum enzyme activity.

The final hydrolysis products of dextran were analyzed using HPLC. Then, 300 µL dextranase was reacted with 900 µL 3% dextran T50 at 40 °C for 0.5 h and 3 h, respectively. Then, the solution was boiled for 5 min to denature dextranase and centrifuged at 10,000 rpm for 10 min. The supernatant was filtered through a 0.45 μm filter. The products were identified and analyzed with the Waters 600 and Waters Sugar-Pak1 (6.5 × 300 mm; Waters, Milford, MA, USA) HPLC with a differential refraction detector. The mobile phase was water at 0.4 mL/min. The column temperature was 75 °C. The injection volume was 20 µL. The standard substances were Glucose (glycarbo#GY1107-500 g 99%), Maltotriose (glycarbo#GY1063-5 g 97%), Maltotetraose (glycarbo#GY1064-50 mg 97%), Maltopentaose (glycarbo#GY1065-50 mg 97%), Maltohexaose (glycarbo#GY1066-50 mg 95%), and Maltoheptaose (glycarbo#GY1067-50 mg 95%), respectively. Quantification was based on calculation of the peak areas. Data acquisition and processing were conducted using Empower GPC software (Waters, Milford, MA, USA).

### 4.8. Effect of Dextranase on Plaque Formation

The effect of dextranase on the biofilms produced by *S. mutans* ATCC 25175 (American Type Culture Collection (ATCC), Manassas, VA, USA) was investigated using the microplate method and SEM. The *S. mutans* were first pre-incubated in a sucrose-free brain heart infusion (BHI) medium at 37 °C for 18 h, and then 600 μL of activated seed solution was inoculated into microplates containing a 1% sucrose BHI medium (Greiner, Frickhausen, Germany). The minimally biofilm inhibitory concentration (MBIC) of the drug was assessed. Microplates were used to measure the biofilm quality of the crystal violet stain in which the biofilm was grown, and then the medium was removed from the microplates containing 1% sucrose BHI medium (contains different concentrations of dextranase) [53]. Then, we slowly added 0.2 mL of a phosphate buffer three times to each well to remove the poorly adsorbed biofilm and medium, and let it dry naturally for 1 h. Then, 0.2 mL of the 0.1% crystal violet solution was slowly added to each well for 5 min, and excess solution was removed and washed three times with a phosphate buffer. After drying, 0.2 mL of 95% ethanol was added to re-dissolve the stained biofilm. Finally, the absorbance at 595 nm was measured on a microplate reader (model 3550; Bio-Rad Laboratories, Richmond, CA, USA). The measurement was repeated three times and averaged, and then we measured the biofilm formation inhibition rate% = (1 − experimental group/control group) × 100 [54].

The sterile slides were placed in a 24-well cell culture plate, and 1 mL of 0.2% sterile gelatin was added for 12 h and then aspirated. The *S. mutans* seed solution and the brain-heart leaching solution (BHI) medium containing 1% sucrose were added to the gelatin-treated 24-well plate in a ratio of 1:9, and the final concentration of the dextranase was 0. 1, 2, 4, 5, 6 U/mL. After anaerobic incubation for 24 h at 37 °C, the slides were removed, gently washed with sterile water, fixed overnight with 2.5% glutaraldehyde, and dehydrated with different concentrations of alcohol gradient. The samples were then dried and sprayed with gold before we observed and detected them by SEM (Model JFC-1600, JSM-6390LA; JEOL, Tokyo, Japan) [43,55].

## 5. Conclusions

In summary, the dextranase-producing MNH15 strain of *C. agarivorans* was screened from marine samples. A higher yield of dextranase was produced when the temperature, initial pH, NaCl concentration, and inducer concentration were 30 °C, 8.0, 5 g/L, and 8 g/L, respectively. The maximum enzyme activity was performed at 40 °C and a pH of 8.0. The dextranase exhibited excellent thermostability and stability at a wide range of pH values (4.0–9.0). The molecular weight of dextranase was around 110 kDa. Sodium fluoride, xylitol, and sodium benzoate, all used in dental care products, had no significant effect on the activity of the dextranase. The result of the HPLC showed that dextran was mainly hydrolyzed to glucose, maltose, and maltoheptaose. Moreover, the dextranase effectively inhibited the formation of biofilm by *S. mutans*. Therefore, dextranase has strong application potential in dental products formulated to remove dental plaque.

## Figures and Tables

**Figure 1 marinedrugs-17-00592-f001:**
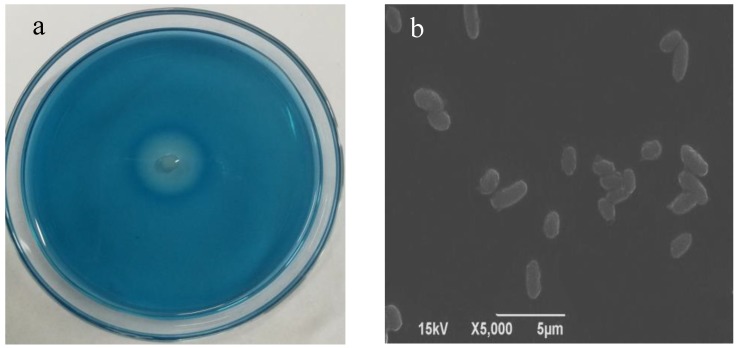
Transparent zone formed by the strain MNH15 on the plate containing blue dextran (**a**) and scanning electron micrograph (**b**).

**Figure 2 marinedrugs-17-00592-f002:**
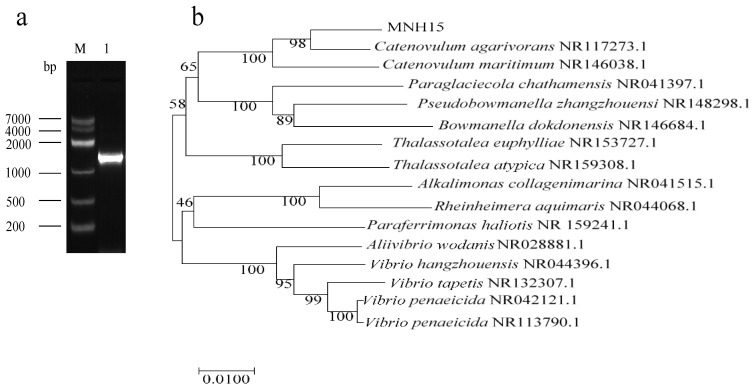
Agarose electrophoresis profiles of 16S rDNA PCR products M: DNA marker (Sangon, China); 1: strain MNH1516S rDNA PCR amplification product (**a**). Phylogenetic tree based on 16S rDNA gene sequences (**b**).

**Figure 3 marinedrugs-17-00592-f003:**
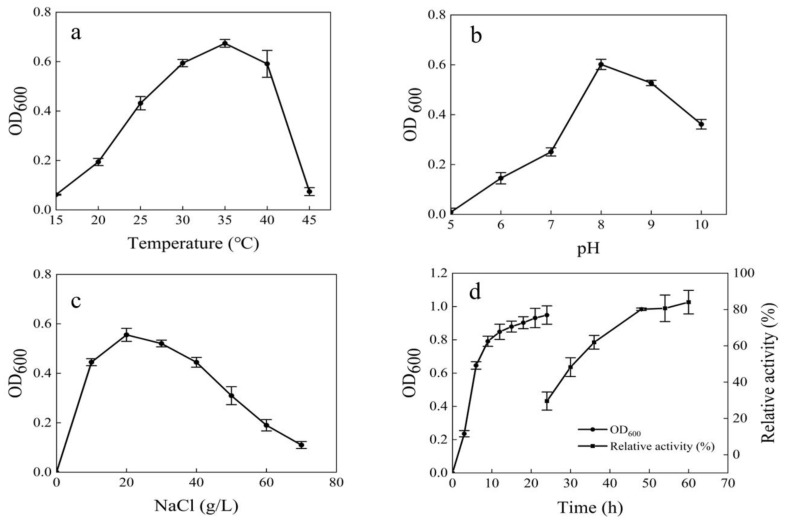
The effects of temperature (**a**), pH (**b**), and NaCl concentration (**c**) on the growth of the strain MNH15 and dextranase producing time (**d**).

**Figure 4 marinedrugs-17-00592-f004:**
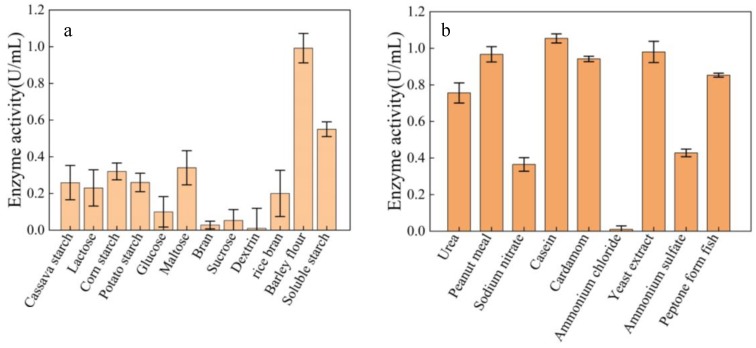
Effect of carbon source (**a**) and nitrogen source (**b**) on dextranase-producing by strain MNH15.

**Figure 5 marinedrugs-17-00592-f005:**
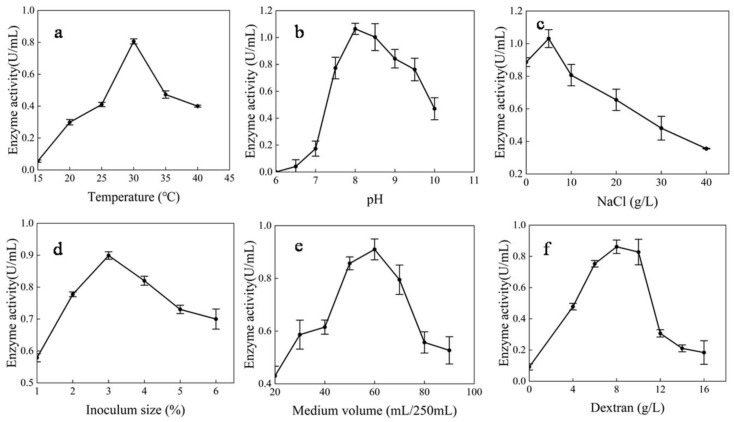
Effects of temperature (**a**), initial pH (**b**), NaCl concentration (**c**), inoculum size (**d**), medium volume (**e**), and inducer concentration (**f**) on dextranase production. For each effect on dextranase activity, the values are shown as percentages of the maximum activities, which were taken as 100%.

**Figure 6 marinedrugs-17-00592-f006:**
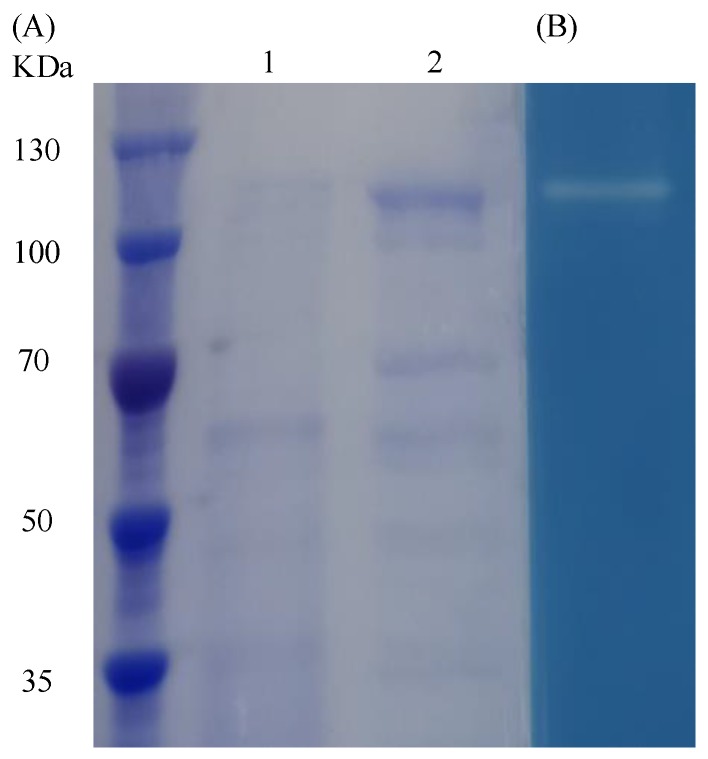
(**A**) Coomassie brilliant blue-stained 8% sodium dodecyl sulfate (SDS)-polyacrylamide gel electrophoresis (PAGE) gel showing the size of the dextranase. M Molecular weight standard (Sangon China, C610011); L1 crude fermentation broth of *Catenovulum agarivorans*. MNH15; L2 the crude fermentation broth of strain MNH15 was concentrated with a 30 K ultrafiltration tube (Millipore). (**B**) Transparent strip showing dextranase activity on an 8% native PAGE gel containing 3% (*w/v*) blue dextran. The numbers on the left indicate the size of the markers.

**Figure 7 marinedrugs-17-00592-f007:**
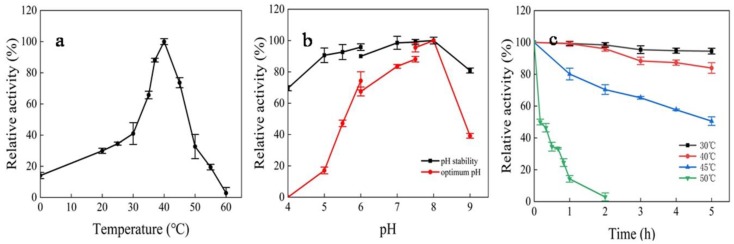
Effects of temperature (**a**) and pH (**b**) on dextranase activity and stability (**c**). For each effect on dextranase activity, the values are shown as percentages of the maximum activities, which were taken as 100%.

**Figure 8 marinedrugs-17-00592-f008:**
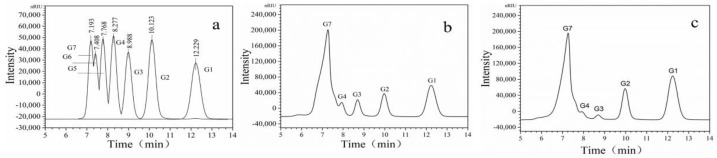
The product of dextranase and 3% dextran T50 reacted at 40 °C for different times was measured by high-performance liquid chromatography (HPLC): (**a**) standard results (G1 to G7) were glucose, maltose, maltotriose, maltotetraose, maltopentaose, maltohexaose, and maltoheptaose standard sugar, (**b**) the results of dextranase hydrolyzed to 3% dextran T50 0.5 h, (**c**) the results of dextranase hydrolyzed to 3% dextran T50 3 h.

**Figure 9 marinedrugs-17-00592-f009:**
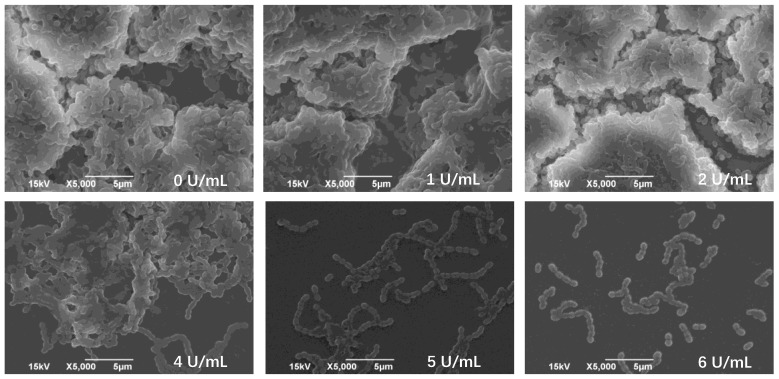
Electron microscopy was used to analyze the effects of different concentrations of dextranase on the biofilm formation of *Streptococcus mutans* on sterile coverslips. Here, 0 U/mL represents a blank control, adding an equal amount of sterile pure water instead of dextranase, and others represent biofilms under the action of 1, 2, 4, 5, and 6 U/mL dextranase, respectively.

**Table 1 marinedrugs-17-00592-t001:** Strains with different hydrolysis abilities.

Isolates	Colony Diameter (d, mm)	Halo Diameter (D, mm)	Hydrolysis Ability (D/d)^2^
MNH10	3.00	5.00	2.78
MNH8	2.00	11.00	30.25
MNH15	2.00	13.00	42.25
BN2	4.00	9.00	5.06
MN7	3.00	6.00	4.00

**Table 2 marinedrugs-17-00592-t002:** Morphological, physiological, and biochemical characteristics of strain MNH15.

Item	Result	Item	Result
Colony color	Milky	Sucrose	+
Shape	Oval	Cellobiose	‒
Gram stain	‒	Hydrogen sulfide	‒
Spore	‒	Mannitol	‒
Motility	+	Glucose	+
4 °C		Arabic candy	‒
37 °C	+	Raffinose	‒
NaCl range for growth (*w*/*v*, %)	0.5–7	Trehalose	+
Arginine decarboxylase	‒	Urea	‒
Lysine decarboxylase	‒	Phosphate	+
Ornithine decarboxylase	‒	Galactose	+
Xylose	‒	Red fresh	+
Lactose	‒	Ribose	+
Maltose	+	Arab alcohol	+

**Table 3 marinedrugs-17-00592-t003:** Effect of metal ions on dextranase.

Reagents	Relative Activity (%) (1 mM)	Relative Activity (%) (5 mM)
Control	100.00 ± 0.07	100.00 ± 0.07
Ca^2+^	95.80 ± 0.56	100.87 ± 0.12
Ba^2+^	97.35 ± 1.10	83.96 ± 0.48
Mg^2+^	88.36 ± 0.86	76.62 ± 2.12
NH^4+^	65.56 ± 1.24	69.37 ± 1.06
Ni^2+^	76.57 ± 0.98	78.81 ± 2.43
Cd^2+^	0.25 ± 0.37	0.00
Fe^3+^	76.62 ± 2.46	0.00
Co^2+^	72.68 ± 2.16	29.14 ± 1.59
Cu^2+^	70.23 ± 2.57	12.35 ± 2.81
Sr^2+^	96.83 ± 2.76	128.71 ± 1.49
K^+^	98.30 ± 2.58	102.10 ± 1.71
Li^+^	0.00	0.00
Zn^2+^	93.64 ± 1.44	86.73 ± 0.81

**Table 4 marinedrugs-17-00592-t004:** Effect of dental caries chemical treatment reagents on dextranase activity. SDS, sodium dodecyl sulfate.

Reagents (*w/v*)	Relative Activity (%)
Control	100.00 ± 0.30
0.1% sodium fluoride	106.64 ± 2.49
0.1% xylitol	103.75 ± 1.23
0.1% sodium benzoate	102.14 ± 1.67
5% ethanol	88.80 ± 3.38
1 mM SDS	52.74 ± 3.05
1% lauric acid	91.20 ± 0.27

**Table 5 marinedrugs-17-00592-t005:** Action of dextranase on diverse carbohydrates.

Substrate	Main Linkages	Relative Activity (%)
DextranT20	α-1,6	73.79 ± 0.52
DextranT40	α-1,6	72.28 ± 0.39
DextranT70	α-1,6	99.56 ± 0.61
DextranT500	α-1,6	100.00 ± 0.25
DextranT2000	α-1,6	95.10 ± 0.72
Soluble starch	α-1,4, α-1,6	4.47 ± 0.49
Pullulan	α-1,4	0.00
Chitin	α-1,4	0.00
Sucrose	α-1,2	0.00
β-cyclodextrin	α-1, 4	0.00
mannan	α-1,4	0.00
Sephadex G-100	α-1,6	82.24 ± 0.27
Sephadex G-200	α-1,6	75.56 ± 0.38

**Table 6 marinedrugs-17-00592-t006:** Proportion of products of hydrolyzed dextran.

Time of Hydrolysis	Hydrolysis Productions (%)				
	Glucose	Maltose	Maltotriose	Maltotetraose	Maltoheptaose
0.5 h	19.16	10.24	6.84	4.96	58.93
3 h	26.06	13.56	2.24	2.83	55.42

**Table 7 marinedrugs-17-00592-t007:** Biofilm inhibitory rates at different concentrations of dextranase.

Addition of Dextranase (U/mL)	Biofilm Formation Inhibition Rate (%)
0	0.00 ± 0.79
1	23.03 ± 1.08
2	35.18 ± 0.85
3	52.30 ± 2.04
4	54.21 ± 1.45
5	74.74 ± 0.74
6	87.12 ± 0.81
7	91.79 ± 0.68

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
