# Peer review of "The Marine *Catenovulum agarivorans* MNH15 and Dextranase: Removing Dental Plaque"

_marinedrugs, 2019, doi:10.3390/md17100592_

Round 1

Reviewer 1 Report

The manuscript entitled "The marine Catenovulum agarivorans MNH15 and the dextranase: removing dental plaque” concerns the discovery of a new enzyme able to hydrolyze α-1,4 glucosidic linkages. Authors report the possible usage of this endo-glycanase in the plaque removal. The topic is interesting. Nevertheless, the manuscript is poor in some parts, as follows:

Authors do not report the purification of the enzyme. If this is true, how can they be sure that the activity will remain the same? Line 187: in my opinion, it is not clear from the TLC which are the real products, since the hydrolysed samples contain spots not aligned with those of the standard. It should be better to analyse the samples with other analytical techniques, such as HPLC, or MALDI-MS. In addition, a quantitative analysis could help in comparison with other dextranases. Line 253: “and it is slightly different compare…” Please explain the differences, since a comparison with a bacterium of the same genus (Catenovulum, ref. 39) is important. What about the toxicity of the dextranase containing sample? To be applied in dental plaque removal, this aspect should be considered. The English language should be revised.

Minor points:

Line 103: tapioca Starch is the same as Cassava starch (figure 4a)? Line 186: no activity. Should be erased There are many references lacking the name of the journal. Only volume and pages have been reported.

Author Response

We have read the comments carefully, and we are really appreciated for your kindly comments. We have revised our manuscript according to the suggestions. The amend parts have been marked as yellow background in the manuscript “The Marine Catenovulum agarivorans MNH15 and the Dextranase: Removing Dental Plaque” (Manuscript ID: Marine Medicine – 594056)

The reviews comments and our responses are as follows:

1.The manuscript entitled "The marine Catenovulum agarivorans MNH15 and the dextranase: removing dental plaque” concerns the discovery of a new enzyme able to hydrolyze α-1,6 glucosidic linkages. Authors report the possible usage of this endo-glycanase in the plaque removal. The topic is interesting. Nevertheless, the manuscript is poor in some parts.

Response: We are appreciated for the reviewer’s comments. We have added some new references in the introduction. Also, we have increased some discussion and the details on the methods parts.   

Authors do not report the purification of the enzyme. If this is true, how can they be sure that the activity will remain the same?

Response: Thank you for the comment. The enzyme was purified. The dextranase is an extracellular enzyme. The broth was centrifugated and the supernatant was ultra-filtered. We have written the details in the methods. In order remain the same level of activity of enzyme, we detected the activity of enzyme before each experiment. For stable of dextranase in 4℃ and room temperature, we have done an experiment. The results showed the dextranase was relative stable.  

Line 187: in my opinion, it is not clear from the TLC which are the real products, since the hydrolysed samples contain spots not aligned with those of the standard. It should be better to analyse the samples with other analytical techniques, such as HPLC, or MALDI-MS. In addition, a quantitative analysis could help in comparison with other dextranases.

Response: Thank you for the comment. we have re-done the experiments, and the spots on the TLC was almost aligned with the standard. We have changed the picture in the manuscript. Some of spots was little bit down compare to the standard, we thought that the reactive solution was more complex than standard solution. The standards solution was added water; however, the reactive solution was added enzyme and buffer. The concentration of enzyme products may be different. We have tried to dilute, but it difficult to see the spots. 

We have sent our samples to analyze by HPLC, but we have received the results yet. We will update the results after we receive.  

Line 253: “and it is slightly different compare…” Please explain the differences, since a comparison with a bacterium of the same genus (Catenovulum, ref. 39) is important.

Response: thank you for the comment. it is the same genus of bacterium,(Catenovulum). The two bacteria were isolated by our lab in different time, however, the molecule weight of two dextranases are much different. Next step, we are detected the gene of the two dextranase and will compare the sequence and the structure of dextranases.

What about the toxicity of the dextranase containing sample? To be applied in dental plaque removal, this aspect should be considered.

Response: Thank you for your comment. It is very important issues about the safety. We have done safety experiments of another bacterium (Arthrobacter oxydans KQ11) that was isolated form marine sample too, and the enzyme solution is safe. Before the application in dental products, the safety experiments, which must be done in authority institutes, will be arranged.  

The English language should be revised.

Response: Thank you for comment. The manuscript has to be English words and grammar checked. And, we have carefully read the manuscript and revised the mistakes. 

Line 103: tapioca Starch is the same as Cassava starch (figure 4a)?

Response: Thank you very much. Tapioca Starch is the same as Cassava starch, and we have revised the mistake.

Line 186: no activity. Should be erased. There are many references lacking the name of the journal. Only volume and pages have been reported.

Response:  Thank you very much. We have revised the mistake. And we have checked the references to revise the mistake.  

Reviewer 2 Report

Generally, in this paper, introduction is not enough. there are many reports about dextranases but, only few papers are introduced.

Methods are also not enough, they should be explained in detail. for example, purification method and SEM observation are not explained in detail. Substrate specificity should be tested using more substrate, not only dextran, but also, other sugar polymers.

In results, productivity should be written as actual activities, not relative values. because the results a and b in Fig 4 can not be compared. yeast extract is not carbon source, it should be nitrogen source.  In fig.5, the intervals are too large, for example, pH interval is 1.0, it should be less than 0.5.

In references, some references are not completed. Author's name must be written in correct, some of them are not enough, furthermore, I think "et al" is italic spelling.   

Author Response

We have read the comments carefully, and we are really appreciated for your kindly comments. We have revised our manuscript according to the suggestions. The amend parts have been marked as yellow background in the manuscript “The Marine Catenovulum agarivorans MNH15 and the Dextranase: Removing Dental Plaque” (Manuscript ID: Marine Medicine – 594056)

The reviews comments and our responses are as follows:

Generally, in this paper, introduction is not enough. there are many reports about dextranases but, only few papers are introduced.

Response: Thank you for your comment. We have revised the introduction, and added references in the introduction.

Methods are also not enough, they should be explained in detail. for example, purification method and SEM observation are not explained in detail.

Response: Thank you for your valuable comments. We have described the experimental methods for enzyme purification and SEM observation in detail. See the revised manuscript for details.

Substrate specificity should be tested using more substrate, not only dextran, but also, other sugar polymers.

Response: Thank you for your comment. We have increased more substrates, and we have added other sugar polymers such as mannan and sucrose. The details were showed in the manuscript (table 5). 

In results, productivity should be written as actual activities, not relative values. because the results a and b in Fig 4 cannot be compared.

Response: Thank you for your comment. We have changed our productivity to actual enzyme activity. Thank you for your valuable advice, which helped us improve our work. See the revised manuscript for details (Fig 4).

 Yeast extract is not carbon source, it should be nitrogen source. 

Response: Thank you for your comment. We have revised the optimization of the production of dextranase by carbon and nitrogen sources. Thank you for reminding us.

In fig.5, the intervals are too large, for example, pH interval is 1.0, it should be less than 0.5.

Response: Thank you for your comment. We have re-done the experiments and the new results were showed in the Fig 5.

7.In references, some references are not completed. Author's name must be written in correct, some of them are not enough, furthermore, I think "et al" is italic spelling.   

Response: Thank you for your comment. we have revised the mistakes in the reference. We managed the reference using EndNote. We find there are mistakes in the reference. Thanks again.

Reviewer 3 Report

Dear Editor,

The manuscript “The Marine Catenovulum agarivorans MNH15 and the Dextranase: Removing Dental Plaque” that submitted to Journal “Marine Drugs” was well described and satisfied. It provides knowledge for medical application of dextranase from marine bacteria, Catenovulum agarivorans.

In my opinion, this manuscript could be published without modifications.

Author Response

We have read the comments carefully, and we are really appreciated for your kindly comments. We have revised our manuscript according to the suggestions. The amend parts have been marked as yellow background in the manuscript “The Marine Catenovulum agarivorans MNH15 and the Dextranase: Removing Dental Plaque” (Manuscript ID: Marine Medicine – 594056)

The reviews comments and our responses are as follows: 

The manuscript “The Marine Catenovulum agarivorans MNH15 and the Dextranase: Removing Dental Plaque” that submitted to Journal “Marine Drugs” was well described and satisfied. It provides knowledge for medical application of dextranase from marine bacteria, Catenovulum agarivorans.

Response: We are very happy to see this anonymous reviewer give praise when reviewing the manuscript, and sincerely thank you for your attention and affirmation of our work.

In my opinion, this manuscript could be published without modifications.

Response: Thank you very much for your comment. Your comment encourages us to do our best in our research.   

Round 2

Reviewer 1 Report

The manuscript entitled "The marine Catenovulum agarivorans MNH15 and the dextranase: removing dental plaque” has been revised, but unfortunately still needs revisions. Authors should primarily improve the English language, and they must answer to point 3 and 4:

Lines from 45 to 48: the 2 sentences must be rewritten Line 57: no “cold-applied enzymes” can be found in reference 27. In addition, what is the meaning of cold-applied? No real chromatographic purification of dextranase has been done. Line 123: why did authors change Yeast extract as best source with barley flour? What about yeast extract? What is the meaning of “The others: potato starch, tapioca starch”? Figure 4a indicates that after barley flour and soluble starch, maltose is a good source. No traces of these comments in the paragraph. Lines 237-238: The sentence “the picture of SEM was clearly observed that…” is wrong

Many other English mistakes are readable all over the manuscript. In addition, misprints such as the usage of capital letters have been found: sentences must start with a capital letter and end with a full stop.

Author Response

Dear Reviewer,

We are really appreciated for your accurate advice about how to improve our manuscript. We have tried our best to revise our manuscript according to the comments, and we have sent the manuscript to the native English speaker to improve language. The revised portions are marked in color in the manuscript.

Responds to the reviewer 1’s comments:

The manuscript entitled "The marine Catenovulum agarivorans MNH15 and the dextranase: removing dental plaque” has been revised, but unfortunately still needs revisions. Authors should primarily improve the English language, and they must answer to point 3 and 4:

Response: Thanks for your comment. We have sent the manuscript to the native English speaker for improving.

Lines from 45 to 48: the 2 sentences must be rewritten

Response: Thanks for your comment. We have rewritten the sentences.

Line 57: no “cold-applied enzymes” can be found in reference 27. In addition, what is the meaning of cold-applied?

Response: Thank you very much for your comments. We are sorry for the misuse of the word “cold-applied enzymes” and the lack of references. We have changed “cold-applied enzymes” to “Cold-adapted enzymes” and added corresponding references.

Generally, an enzyme having an optimum catalytic temperature of about 30 ℃ and maintaining a certain catalytic efficiency at 0 ℃ is called a low temperature enzyme, and is also called a cold-adapted enzymes[1].

[1]. Margesin, R.; Schinner, F., Properties of cold-adapted microorganisms and their potential role in biotechnology. Biotechnology - J BIOTECHNOL 1994, 33, 1-14.

No real chromatographic purification of dextranase has been done

Response: Thank you very much for your comment. Yes, we did not purify the dextranase with chromatograph. We purified the dextranase with different filters. We cannot say the dextranase is pure. It is primary pure after filter.

We have finished the analysis of products. The detailed revisions respectively are:

High performance liquid chromatography (HPLC) showed that maltoheptaose, glucose and maltose were the main hydrolysates of dextranase (Figure 9). The peak area of the hydrolyzate detected by HPLC was quantified using Empower GPC software (Table 6). The results showed that maltoheptaose and maltose had no increasing trend compared with glucose over time. In addition, when the hydrolysis reaction time was extended from 0.5 h to 3 h, the amount of maltotriose and maltotetraose decreased slightly, which was similar to the results reported in the literature.

Fig. 9. The product of dextranase and 3% dextran T50 reacted at 40 ℃ for different times was measured by HPLC:

a: standard results (G1 to G7 are glucose, maltose, maltotriose, maltotetraose, maltopentaose, maltohexaose, maltoheptaose standard sugar); b, c was the results of dextranase hydrolyzed 3 % dextran T50 0.5 h, 3 h.

Table 6. Content of sugar (%) in dextran hydrolysate.

Time of Hydrolysis

Hydrolysis Productions (%)

Glucose  Maltose  Maltotriose Maltotetraose  Maltoheptaose

0.5h          19.16    10.24    6.84      4.96          58.93

3h           26.06    13.56    2.24      2.83          55.42

Line 123: why did authors change Yeast extract as best source with barley flour? What about yeast extract?

Response: Thank you very much for your comments. According to the comments of reviewers. We have done the carbon and nitrogen source optimization experiments again and the Yeast extract has been listed in the nitrogen source. The barley flour, rice bran and soluble starch as carbon sources in the new experiments.

What is the meaning of “The others: potato starch, tapioca starch”?

Response: Thanks for your comment. the starch was made of potato or tapioca. Starch is mainly carbon sources that was used in fermentation. We chose different starch that was made from different plants [2-5].   

[2]. Yoo, S. H., & Jane, J. (2002). Molecular weights and gyration radii of amylopectins determined by high performance size-exclusion chromatography equipped with multi angle laser-light scattering and refractive index detectors. Carbohydrate Polymers, 49, 307–314.

[3]. Hizukuri, S. (1996). Starch: Analytical aspects. In A. C. Eliasson (Ed.), Carbohydrates in food (pp. 347–429). New York, NY: Marcel Dekker.

[4]. Vermeylen R, Goderis B, Delcour J A. An X-ray study of hydrothermally treated potato starch[J]. Carbohydrate Polymers, 2006, 64(2):364-375.

[5]. Zhu F. Composition, structure, physicochemical properties, and modifications of cassava

starch. Carbohyd Polym.2015;122:456-80.

Figure 4a indicates that after barley flour and soluble starch, maltose is a good source. No traces of these comments in the paragraph

Response: Thanks for your comment. We have revised in the manuscript.

The detailed revisions respectively are:

Barley flour was the best carbon source for the production of dextranase. Soluble starch and maltose have a slight effect on the strain producing dextranase. Potato starch, tapioca starch, lactose, corn starch, glucose, rice bran, bran, sucrose and dextrin cannot promote the production of dextranase.

Lines 237-238: The sentence “the picture of SEM was clearly observed that…” is wrong

Response: Thank you very much for your comments. We have revised the sentence.

The detailed revisions respectively are:

When the concentration of dextranase was continuously increased, the adhesion of bacteria decreased, the thickness of the biofilm gradually become thinner, and the structure becomes loosed.

Many other English mistakes are readable all over the manuscript.

Response: Thank you very much for your comments. We have sent the manuscript to  

In addition, misprints such as the usage of capital letters have been found: sentences must start with a capital letter and end with a full stop.

Response: Thank you very much for your comments and suggestions. We have carefully revised the manuscript.

Yours sincerely,

Mingsheng Lyu

Reviewer 2 Report

I donut think it is enough. Because some results and discussion will be needed. Please see the comment attached.

Author Response

Dear Reviewer,

We are really appreciated for your accurate advice about how to improve our manuscript. We have tried our best to revise our manuscript according to the comments, and we have sent the manuscript to the native English speaker to improve language. The revised portions are marked in color in the manuscript. The main corrections in the paper and the responds to the reviewer’s comments are as flowing:

Responds to the reviewer 2’s comments:

I think this enzyme may not be useful for dental washes, because it does not mention about resistance to surfactants and detergents.

Response: Thank you very much for your comments. In table 4, we have done experiments about resistance to the chemicals which used in mouthwash or toothpaste normally. The results showed the dextranase could maintain the activity.

If you hope its possibility would be shown, the resistance should be demonstrated on the dextanase activity and stability, for example, to sodium dodecyl sulfonate (SDS), and fatty acids,like a lauric acid.

Response: Thanks for your comment. We have added experimental data based on your suggestions. we found that 1% lauric acid could decrease 8.8% dextranase activity. 1 mM sodium dodecyl sulfate (SDS) could decrease 47.3% dextranase activity.

The detailed revisions respectively are:

 Effect of chemical treatment reagents on dextranase activity

1%lauric acid

91.20±0.27

1mM SDS

52.74±3.05

Also, We detected the effect of 1% lauric acid and 1 mM SDS on the stability of dextranase at room temperature and 4℃, respectively. The results showed that dextranase was added to 1% lauric acid. As time goes on, the dextranase activity kept relatively stable.

Effect of 1% lauric acid and 1 mM (a) sodium dodecyl sulfate (SDS) (b) on the stability of dextranase. (The values are shown as percentages. The maximum activities were set as 100 %.)

3.In the figure 7, titles should be changed into Effects of pH and temperature on dextranase results, not only optimum condition.

Response: Thanks for your kindness comment. We have changed the title of Figure 7 to "Effects of temperature(a) and pH(b) on dextranase activity and stability (c)".

We would like to express our great appreciation again for the comments which help us to improve our manuscript significantly.  

Yours sincerely,

Mingsheng Lyu

Round 3

Reviewer 1 Report

Please change in Table 6 the following: maltoheptaose instead of maltopentaose

Author Response

Dear Reviewer,

We are really appreciated for your kindly help. Your comments has made us improve our manuscript better and better. Thank you very much.

We have changed the Table 6: maltoheptaose instead of maltopentaose. And we have changed the maltopentaose to maltoheptaose in the Abstract and Conclusions. All the changes have been set yellow background.

We would like to express our great appreciation again for your help.  

Yours sincerely,

Mingsheng Lyu

Reviewer 2 Report

It is good to revise and add the data.

it is ok for publication of this article.

Author Response

Dear Reviewer,

We would like to express our great appreciation for your kindly help. Your comments has made us improve our manuscript better and better. Thank you very much.

Yours sincerely,

Mingsheng Lyu